# Smart Steering Sleeve (S^3^): A Non-Intrusive and Integrative Sensing Platform for Driver Physiological Monitoring

**DOI:** 10.3390/s22197296

**Published:** 2022-09-26

**Authors:** Chuwei Ye, Wen Li, Zhaojian Li, Gopi Maguluri, John Grimble, Joshua Bonatt, Jacob Miske, Nicusor Iftimia, Shaoting Lin, Michele Grimm

**Affiliations:** 1Department of Mechanical Engineering, Michigan State University, East Lansing, MI 48824, USA; 2Department of Electrical and Computer Engineering, Michigan State University, East Lansing, MI 48824, USA; 3Physical Sciences Inc., Boston, MA 01810, USA

**Keywords:** non-intrusive driver monitoring, flexible sensor, sensor integration

## Abstract

Driving is a ubiquitous activity that requires both motor skills and cognitive focus. These aspects become more problematic for some seniors, who have underlining medical conditions and tend to lose some of these capabilities. Therefore, driving can be used as a controlled environment for the frequent, non-intrusive monitoring of bio-physical and cognitive status within drivers. Such information can then be utilized for enhanced assistive vehicle controls and/or driver health monitoring. In this paper, we present a novel multi-modal smart steering sleeve (S${}^{3}$) system with an integrated sensing platform that can non-intrusively and continuously measure a driver’s physiological signals, including electrodermal activity (EDA), electromyography (EMG), and hand pressure. The sensor suite was developed by combining low-cost interdigitated electrodes with a piezoresistive force sensor on a single, flexible polymer substrate. Comprehensive characterizations on the sensing modalities were performed with promising results demonstrated. The sweat-sensing unit (SSU) for EDA monitoring works under a 100 Hz alternative current (AC) source. The EMG signal acquired by the EMG-sensing unit (EMGSU) was amplified to within 5 V. The force-sensing unit (FSU) for hand pressure detection has a range of 25 N. This flexible sensor was mounted on an off-the-shelf steering wheel sleeve, making it an add-on system that can be installed on any existing vehicles for convenient and wide-coverage driver monitoring. A cloud-based communication scheme was developed for the ease of data collection and analysis. Sensing platform development, performance, and limitations, as well as other potential applications, are discussed in detail in this paper.

## 1. Introduction

Driving is an omnipresent everyday activity, and it is estimated that U.S. drivers spend an average of more than 50 min behind the wheel every day [1]. Simultaneously, driving is a complex task that requires the integration of sensory, motor, and judgment skills. Therefore, it provides a unique controlled environment that can be used to monitor the driver’s physiological and psychological state, which can be seamlessly integrated into driver assistance systems such as driver fatigue/distraction warning [2,3] and lane keeping control [4,5] techniques. In addition, it can be used for the continuous monitoring of a driver’s health and cognitive capability [6]. Especially for elder drivers with potential cognitive and physiological impairments, it is important to monitor their health conditions to support maintaining a safe driving experience.

Various biomarkers and physiological measurements have been proposed as objective metrics to evaluate the physical and cognitive state of human beings, including electrocardiogram (ECG), electroencephalogram (EEG), electromyography (EMG), electrodermal activity (EDA), photoplethysmography (PPG), and skin temperature. The ECG, EEG, and EMG signals are measurements used to evaluate and monitor the function of the heart [7,8], brain [9,10], and muscles [11,12], respectively. EDA measures the variation in the skin conductivity caused by sweat secretion [13,14,15], whereas PPG is a measure used to detect the change in the blood volume using optical devices [16,17].

Several sensing systems have been developed for driver monitoring using the aforementioned physiological markers over the past decade. Lee et al. created a sensors-gathered device to measure a driver’s EMG and PPG in real time to estimate their emotional response [18]. Said et al. designed a wearable bracelet embedded with EMG, ECG, and EDA bio-sensors to detect abnormal bio-psychological parameters that may result in an emergency situation, though with a short delay [19]. Lee et al. developed a driving safety system with bio-sensors to monitor the excitability and fatigue of the driver in real time, and telematics technologies were used to transmit the data to the nearest medical service centers [20]. In addition, Li et al. prototyped a system with PPG sensors to measure the heart rate variability (HRV) and then transmit the signals via Bluetooth technologies, followed by wavelet analysis processing for continuous driver monitoring [21].

Despite advances in the development of wearable sensors such as smart watches (e.g., Apple Watch) and wrist bands (e.g., Fitbit) for driver physiological measurements, it is not easy to achieve a high adoption rate when their use is not compulsory. Toward that end, several unobtrusive systems have also been developed to measure ECG [22,23], EEG [24], and HRV signals [25]. Visual signals and video data have also been proposed to monitor the health conditions of drivers [26,27,28]. However, an integrated system used to continuously and conveniently monitor a combination of the physiological signals is still lacking. To address the challenges, we developed a novel multi-modal smart steering sleeve (S${}^{3}$) system with an integrated flexible sensing platform that can continuously and conveniently measure the driver’s physiological markers. The sensor is integrated into a flexible tape, which is then affixed on top of a steering wheel sleeve (see Figure 1a). As the driver needs to constantly hold and move the steering wheel, the developed platform provides a convenient and robust method for driver monitoring without the need to remember to charge or wear any devices.

More specifically, the integrated sensor comprises three measurement modules: a force-sensing unit (FSU), a sweat-sensing unit (SSU), and an EMG-sensing unit (EMGSU). The FSU uses a piezoresistive material, and the sensitivity is uniform along its length. The SSU is designed with interdigitated electrodes, providing great linearity between its resistance and time-evolving sweatiness. The EMGSU consists of two active electrodes and one reference electrode distributed along the sides of the SSU. The EMGSU can acquire the EMG signals effectively with the aid of an instrumental amplifier. The three sensing modules (i.e., FSU, SSU, and EMGSU) are all made from film-shaped flexible materials and tightly integrated into a tape-like strip attached to an off-the-shelf steering sleeve. Data acquisition circuits were designed to retrieve the data, and a cloud-based communication architecture was developed to send relevant data to a cloud database for ease of data management and analysis.

The contributions of the paper include the following: first, a novel multi-modal smart steering sleeve system was developed that can conveniently and continuously monitor driver physiological conditions through grip force, palm sweat, and EMG. A cloud communication scheme was also developed to facilitate data management and analysis. Second, mechanisms and devices were designed and developed for an accurate and robust characterization of the three sensing units. Third, comprehensive characterizations were performed on the integrated sensor platform, with promising results demonstrated. Finally, various potential applications of the developed sensors are discussed. The designed prototype is compact in size, simple in structure, and low in cost, and the integrated sensor can robustly measure the aforementioned three physiological indicators. The developed sensor suite is a key enabler for our forthcoming study to correlate physiological signals with driving quality (in varying scenarios) and cognitive capabilities, the investigation of which will be reported in our future publications. Variations among individuals and different driving conditions will be addressed through the novel design of machine learning algorithms.

The remainder of the paper is organized as follows. Section 2 introduces the design and mechanisms of the integrated sensor and the measurement system. Section 3 presents the experimental setup and circuits for the sensing units, whereas Section 4 displays the characterization results of the integrated sensor. Section 5 discusses the key findings, limitations, and future work. Finally, conclusions are drawn in Section 6.

## 2. Sensor Design and Mechanism

### 2.1. System Overview

The developed cloud-based smart steering sleeve with multi-modal sensing capabilities is illustrated in Figure 1a. It features an add-on steering sleeve with an easy-to-integrate sensing platform that can non-intrusively and continuously measure a driver’s physical and physiological signals, including EDA, EMG, and hand pressure. Besides individual driver-based analysis, data from multiple drivers are sent to a secure cloud database, where comprehensive data analytics can be performed on the multi-level, multi-modal, and time-series datasets to identify patterns of correlation between behavior and the signals.

A prototype of the flexible sensor is shown in Figure 1b. All sensing modules were made from flexible film materials. Specifically, the FSU was made from a piezoresistive film and copper foils, which were used as the sensing element and electrodes, respectively. The SSU and EMGSU were made from copper electrodes patterned on a single polyimide substrate. The sensing units were tightly integrated in a multi-layer structure, where the SSU and the EMGSU were integrated in one sheet and placed on the top so that it directly contacted the driver’s skin, and the FSU was placed underneath.

The diagram of the measurement system based on the integrated sensor is shown in Figure 2a, and the prototype of the data acquisition circuits is depicted in Figure 2b. Voltage driver circuits were applied towards the FSU and the SSU to capture their resistance changes. We used direct current (DC) as the power supply of the FSU voltage divider, whereas alternative current (AC) was exploited for the SSU power supply to eliminate the crosstalk between these two sensing units. The EMG signals are relatively weak, so an instrumental amplifier was applied for the EMGSU. We used MCP3008 (Microchip Technology, Chandler, AZ, USA) as the analog-to-digital converter (ADC) chip, and Pi pico (Raspberry Pi) as the microcontroller unit (MCU). The communication between the ADC and the MCU was realized through the serial peripheral interface (SPI). The acquired data were transmitted to a central processor via a Bluetooth module HC-05 with the serial communication protocol. A LabVIEW interface was developed in the central processor to display and save the data. At the same time, the computer could use the hot spot enabled by a smart phone to transmit the data to the cloud in real time. At a 5 Hz data uploading frequency, a 140 Byte/s transmission speed could be achieved. The cloud communication was realized by using an encrypted secure Google cloud database. The Google drive provides a convenient method for the rapid transfer of data, adopting a command line utility called “gdrive” that is specific to a dedicated subject with a specific risk score measurement account, allowing for the separation of user data and predictions.

### 2.2. FSU Design and Mechanism

The FSU was constructed by sandwiching a piezoresistive film (“Velostat” or “Linqstat”) between two electrode strips as shown in Figure 3a. The electrode strips were made from copper adhesive foils (3M, Saint Paul, MN, USA) with a sheet resistance of less than 0.005 $\Omega /{\mathrm{inch}}^{2}$. Within the sensor, the copper films were 0.07 mm in thickness and were cut into two different widths, 25.40 mm and 6.35 mm. The length of the electrode strips can vary depending on the circumference of the steering wheel to be covered. For sensor assembly, the bottom electrode strip was firmly adhered to the piezoresistive film, while a cross-shape pattern of copper protrusions (Figure 3b) was attached to the upper electrode strip with air gaps created between the upper electrode and the piezoresistive film. This configuration was specifically designed to improve the sensitivity, which is attributed to both the piezoresistive effect and the sensitive contact area of the upper electrode. During the sensor operation, when no external force is applied on the upper copper film, sensitive contact points are made only between the protrusions and the piezoresistive film. With applied external force, additional contact points will be generated across the air gaps due to the deformation of the upper electrode, and the sensitive contact area increases as the external force increases.

It is to be noted that the area and height of the air gap are the two key parameters critical to the sensitivity and operational range of the FSU. Air gaps with a larger area and lower profile are expected to enable a higher sensitivity but have a limited dynamic range because all of the gaps can easily collapse with a small external force. On the other hand, if the gap is too large, it will be hard to generate contact points even with a large force because of the stiffness of the piezoresistive film and the protrusion parts of the upper electrode. Moreover, electrodes of excessive thickness will make the system bulky, and hence less appealing to our target application. In the current design, we fixed the areal dimension of the air gap to be 40 mm${}^{2}$ and tuned the height from 0.07 mm to 0.14 mm to 0.21 mm to study the impact of the gap height on the sensitivity and dynamic range of the FSU.

### 2.3. SSU Design and Mechanism

As shown in Figure 4a, the SSU utilized an interdigitated electrode configuration. The length and width of the interdigitated finger-shaped electrodes were 4.2 mm and 1 mm, respectively, and the distance between the adjacent finger-shaped electrodes was 1 mm. When held by the driver, the sweat in the palm will infiltrate through the sensor, resulting in a decrease in resistance between the two electrode terminals. A larger amount of sweat will short more interdigitated electrode fingers. Therefore, the resistance between the two electrode terminals can be continuously monitored as the volume or mass of the sweat changes over time. The SSU was built using a carbon mask printed on a circuit board thermal transfer paper using an ink printer, and was then transferred to a copper-clad polyimide film (DuPont, Wilmington, DE, USA) using a hotplate at 170 °C. After that, the copper-clad polyimide film was submerged in a ferric chloride copper etching solution to remove unwanted copper. Finally, the thermal transferred mask was removed using acetone followed by methanol and a deionized (DI) water rinse.

### 2.4. EMGSU Design and Mechanism

An EMG signal is relatively weak when measured from the hands/palms. In our design shown in Figure 4b, the two active electrodes were connected to the two differential terminals of the instrumental amplifier. As a result, the common mode interference between the two active electrodes is attenuated effectively. The reference electrode was connected to the middle resistance point between the two terminals, which determines the gain of the amplifier. To limit the common mode interference, the distance between the active electrodes and the reference electrode must be over a minimum value. Hence, they were separately placed on the left and right sides of the SSU. Moreover, considering that a reliable reference should be assured to effectively obtain the EMG signals, the reference electrode was designed to be wider than the active electrodes. The EMG electrodes were constructed together with the SSU using the previously described method.

## 3. Methods for Sensor Characterization

### 3.1. FSU characterization Setup

To calibrate the FSU performance, including the range, sensitivity, and hysteresis, a load cell (Alamscn) was used to measure the force applied to the sensor as shown in Figure 5. The load cell was based on strain gauges connected as a Wheatstone bridge, and its sensitivity depended on the voltage provided to the load cell, which was measured before each calibration. The output voltage of the load cell was amplified by an instrumental amplifier INA128 (Texas Instruments, Dallas, TX, USA ), and was measured by a VICTOR 89B multimeter (Victor Instruments, Bunsen Irvine CA, USA). As the FSU is resistance-based, a voltage divider circuit was used to measure the varying resistance by Ohm’s law. The power supply of the voltage divider was a 3.3 V source generated by 6 V Ni-MH batteries and a voltage regulator (LD1117V33, STMicroelectronics, Geneva, Switzerland). A 47 Ω resistor was connected in series with the FSU. The sensitivity curve of the FSU could then be obtained by calibration, which is provided in Section 4.1.

During the calibration, the FSU was placed on an 80 mm long plastic bar with a curved hollow inner surface, the radius of which was 20 mm, to simulate the radius of the cross-section of a steering wheel. Another plastic bar was instrumented to press the FSU for loading, and the two bars were coupled on the curved surface. To simulate the conditions where the force is generated between the palm and steering wheel cover, and to evenly distributed the force, two pieces of rubber were attached to the two bars on the curved surface. Three bolts with ball heads were applied to press on the FSU, and they were mounted on an aluminum plate fixed on the frame above the FSU. When screwed, the bolts pushed the aluminum plate and generated a continuously varying force on the FSU. The load cell, with a range of 0 to 100 N, was fixed at one end of the frame as the standard device to measure the applied force.

### 3.2. SSU Characterization Setup

To obtain the characteristics of the SSU, normal saline (Nurse Assist, Inc., Haltom City, TX, USA) was used to simulate the humidity caused by sweat. During testing, the SSU was covered by a non-woven cotton gauze (Dealmed, Brooklyn, NY, USA) to distribute the saline evenly on the surface. Saline (1 mL) was dropped evenly on the gauze to simulate the sweating condition of the driver’s palm. After saline infiltrated the gauze, a 40 W heat lamp (General Electric, Boston, MA, USA) was used to heat up and vaporize the applied saline in order to simulate sweat evaporation and decreased humidity. A 1-kΩ resistor was connected in series with the SSU to make a voltage divider circuit, and the output resistance of the SSU was measured over time. To monitor the evaporation rate of saline, a digital balance with 0.01 g resolution was used to measure the mass of the remaining saline on the gauze. The experimental setup is illustrated in Figure 6.

### 3.3. EMGSU Experiments

To evaluate the effectiveness of the EMGSU design, the EMGSU was placed on the steering wheel (Logitech Inc., Lausanne, Switzerland) and the driver’s hand gripped the sensor with different forces, as illustrated in Figure 2b. Then, the EMG signal from the hand was amplified by the commercial amplifier and acquired by a data acquisition (DAQ) device (USB6008, National Instruments, Austin, TX, USA) with a 10 kHz sampling frequency and a ±10 V input range. The acquired signals were displayed and saved by software developed in LabVIEW (National Instruments, Austin, TX, USA).

## 4. Characterization Results

### 4.1. FSU Characteristics

Figure 7 showed the characterization curves of the FSU with different air gap heights before and after applying the forces. The x-axis is the force, and the y-axis is the ratio of the voltage divider output (Vo) to the power supply voltage (Vcc). The FSUs with different chamber thicknesses showed similar sensing ranges, with a maximum detectable loading force of around 25 N. However, the hysteresis curves of the FSUs showed a significant difference. In this case, the hysteresis is the measurement of the maximum difference in *Y* offset between the loading and unloading curves of the same sensor under the same loading force. Hence, the hysteresis of the sensitivity curves can be derived as [29]:(1)$$Hysteresis\%=|\frac{Y_{mu}-Y_{ml}}{Y_{max}-Y_{min}}|\cdot 100\%$$
where the $Y_{mu}$ and the $Y_{ml}$ are the values in the *Y* axis, where the maximum difference occurs in the unloading and loading curves, respectively. The $Y_{max}$ and the $Y_{min}$ are the maximum and the minimum values in the range of the *Y* axis, respectively.

The results show that the hysteresis was 53.646%, 19.298%, and 5.172% for FSUs with air gaps of 0.07 mm, 0.14 mm, and 0.21 mm, respectively. The hysteresis was reduced as the air gap height increased, perhaps due to the increased rigidity of the upper electrode with thicker protrusions. Because of the small hysteresis, we chose the force sensor with the 0.21 mm air gap to be applied in the integrated sensor.

In practice, the FSU should be long enough to cover the circumference of the steering wheel, so the uniformity of the sensor performance along the length needs to be carefully assessed. As such, the 300 mm long FSU with an air gap of 0.21 mm height was calibrated at three different spots—left, middle, and right—as illustrated in Figure 8a, where the corresponding characterization curves are shown in Figure 8b–d. It can be seen from Figure 8 that the sensitivity of the FSU can be treated as piece-wise linear and polynomial, and the data points can be divided into two parts: one part is linear between 0 N and either 5 N or 10 N, and the other part from the end of the first part to 25 N, which has a third order polynomial relationship. The reason for the piece-wise characterization is that, as we increased the load before the air gap in the FSU disappeared, the resistance decreased rapidly and the sensitivity/slope was relatively high. Conversely, after the air gap disappeared, it was harder to generate more connecting points between the upper electrode and the piezoresistive film. Thus, the resistance of the FSU consequently decreased more slowly while the pressure increased, resulting in a lower sensitivity. The output ranges of the three regions of the FSU were similar, in that they are all approximately within 60% of the voltage of the power supply (Vcc). In the process of measurement, the applied force can be calculated by determining the contacting area and applying the appropriate force–voltage relationship.

The comparison of our FSU against the commercial load cell (TAL220 Sparkfun Electronics, Niwot, CO, USA) is illustrated in Figure 9, with the data from six repeated experiments, showing that the FSU measurements match the results obtained from the standard load cell with small deviations.

### 4.2. SSU Characterization

The relationship between the mass of the saline and the resistance of the SSU is shown in Figure 10. The data are from six repeated experiments. Note that, as the mass variation of sweat secreted by a driver is expected to be less than 1.2 mg as a result of short-term mental stress [30], the range of volumes of saline used in our calibration is sufficient to estimate this relationship, even taking the accumulation of the sweat into consideration [31]. The results in Figure 10 show that the relationship is almost linear.

However, there is an important observation that should be taken into consideration for the integrated sensor. As the force applied on the SSU increases in driving activities—for example, when the driver is holding the steering wheel—the resistance of the SSU decreases. The reason is that the substrate material and the copper layer of the SSU are hydrophobic; therefore, when pressed, the sweat droplets will be compressed. This will result in a larger contact area between the SSU and the palm, covering more interdigitated electrode figures, and thus decreasing the measured resistance. In other words, there is a crosstalk between the SSU and FSU. According to S. Yao et al. [32], the impedance between the tissue in the palm and the electrodes of the SSU can be measured under an AC source. The stimulation from the varying force is at a low frequency, and if a relatively higher frequency AC source is applied to the SSU, filters with an appropriated cut-off frequency can be used to suppress the influence of the varying force.

Therefore, an AC source with a 2 Vpp voltage and 100 Hz frequency was applied using a direct digital synthesizer (DDS) module based on a field programmable gate array (FPGA) chip. During the test, saline was added gradually, decreasing the impedance between the palm and the SSU. At the same time, the applied holding force was slowly changed by applying pressure through the hand. In Figure 11a, it can be observed that the varying force can still result in ripples in the resistance of the SSU, but the baseline of the resistance is consistent with the process of adding saline. After a fifth-order Butterworth filter with a high cut frequency of 10 Hz was applied, as in Figure 11b, the ripples were smoothed out, leaving only the baseline that can be used to represent the varying mass of the sweat.

### 4.3. EMGSU Characterization

The amplification analog circuit for the EMG signal acquisition was based on an instrumental amplifier. The reference electrode connected the middle resistance point of the two gain-determining pins of the instrumental amplifier and the palm skin through a driven right leg circuit [33], and the two active electrodes were connected to the two differential input pins of the amplifier. The reference pin of the amplifier was connected to the output of a 2.5 V regulator, so the baseline of the amplified signal was 2.5 V. This can be acquired by the analog-to-digital converter (ADC) chip without the negative voltage conversion function. During the characterization, the USB6008 data acquisition (DAQ) device was used, and the amplified signals were obtained as shown in Figure 12. It is obvious that the EMG signals were obtained successfully.

However, there was still crosstalk between the EMGSU and the SSU if the SSU used an AC source. Because of the conductivity of the palm skin and tissue under the skin, the 100 Hz AC voltage resulted in an oscillation of the output of the EMGSU at the same frequency (Figure 13a). This issue could be resolved by turning off the AC source of the SSU when selecting the EMG channel as shown in Figure 13b.

For the EMGSU, the commercialized amplifiers were integrated, and a commercial EMG sensor (EMG Detector, Seeed Studio, China ) was used to validate the accuracy of the EMGSU. The comparison between the EMGSU and the commercial EMG sensor is illustrated in Figure 14, which shows a high correlation with the commercial EMG sensor.

## 5. Discussion

The design of the integrated sensor is compact in size, flexible in configuration, multi-modal for various functions, and easy to deploy. The simple structure and mechanism also make it low-cost and convenient to build. This provides a unique platform that can continuously and conveniently monitor a driver’s physiological status, without the need for heavy and unwieldy wearable sensors. The results show that the three sensing units in the integrated sensor were successfully developed to achieve desired functionalities. As validated against a commercial load cell and a commercial EMG sensor, the accuracy of the FSU and EMGSU can be confirmed. With the designed simple structure and sensing principle of the SSU, which has great repeatability, the SSU can also function consistently under variations in humidity due to the presence of sweat.

Similar to most commercial products, the FSU does not have a linear relationship between the input force and the output voltage under the use of a voltage divider, and this is mainly because of the nonlinearity between the force and resistance of the FSU. However, the sensitivity can be regarded as a piece-wise function, combining a linear region and a third-order polynomial relationship. For example, in the range of 0 to 7 N, the linear fit can be used, and between 7 N and 25 N, the third-order polynomial fit can describe the relationship. This phenomenon inspires us to seek an approach to improve the linear characteristics of the FSU. For example, the protrusions of the upper electrodes can be designed in a different pattern. Nevertheless, the consistency of the sensitivity curves among different parts of the FSU confirms that the driver’s holding force can be monitored with little bias no matter which part of the FSU is pressed. Although the sensing range of the FSU is relatively limited, it will still be able to cover the gripping force of senior drivers under normal driving conditions, which is the main objective of the ongoing study.

The SSU results show good linearity between the mass of the sweat and the resistance of the SSU in a wide range. Hence, the resistance between the two electrodes of the SSU can represent the driver’s sweatiness quantitatively. The 100 Hz AC source of the voltage divider for the SSU can eliminate the crosstalk from the FSU, but it requires a high performance from the DAQ devices, especially in the sampling rate. Although The USB6008 DAQ device is sufficient for the measurement tasks of the current SSU design, an MCU with the ADC function and a sampling frequency of more than 5 kHz is recommended as an alternative choice when taking monetary cost into consideration.

As for the EMGSU, the results of the acquired signals show a successful design but there are still issues that need to be further addressed. For example, when the attachment between the palm skin and the EMGSU is not firm and reliable, the EMGSU and the amplifier circuit will absorb the 60 Hz interference from the surrounding environment, resulting in oscillation as the output. Applying an electromagnetic shield to cover the amplifier might be a good solution to partially solve this problem. Moreover, it will be better to develop an oscillation detection algorithm in the MCU or software programs so that the environmental interference can be removed in all situations.

In the S${}^{3}$ system, the sensing units of the FSU, SSU, and EMGSU are all fabricated with copper foil and degradable polymer membranes, which are non-toxic and environmentally friendly materials. Moreover, the circuits serving the S${}^{3}$ system all work under low-power and low-frequency conditions. Therefore, there will be no harmful effect on human health or the environment caused by the sensing platform.

The $S^{3}$ system will have potential applications in many different fields. First of all, in conjunction with an eye-tracking system, the system can be used to monitor fatigued and/or distracted drivers, and driver assistance system (e.g., driver alert, assistive control) can then be integrated to enhance driver safety. Second, as the $S^{3}$ system can continuously monitor the EMG and palm force of the driver, it can potentially be used as a platform for the detection of Parkinson’s disease and epilepsy. Last but not least, the $S^{3}$ system can be integrated with an onboard drive quality monitoring system so that the subject’s driving quality in terms of metrics (including lane deviation and distances to surrounding vehicles) can be synchronously monitored and evaluated, providing crucial contextual information about the physiological measurements in the controlled, cognition-involved driving environment. This system can then potentially be used to evaluate the cognitive capability of drivers. We are developing such a system and plan to perform a subject test with normal seniors and cognitively impaired patients on a driver simulator, where the developed $S^{3}$ sensing platform is a key enabler. The study will investigate the correlation between the driver’s physiological signals and their driving quality and cognitive capabilities, which will be reported in our future publications. Variations among individuals and under different driving conditions will be addressed through novel designs of machine learning algorithms.

## 6. Conclusions

Driving is a ubiquitous activity in daily life that requires driver’s motor skills and cognitive focus. Therefore, a non-intrusive sensing platform for monitoring both the physiological and psychological status of the driver can be seamlessly integrated into driving assistance systems to improve the safety of the driving experience. We designed and developed an integrated, cloud-based multi-modal sensor with force, sweat, and EMG-sensing capabilities. The three sensing modules were all made from flexible film materials, making them easy to integrate on a steering wheel sleeve. The FSU was fabricated with piezoresistive films and used copper foil with protrusions as its electrodes, achieving an evenly distributed sensitivity within a measurement range of 25 N, and with an output range within 60% of the power supply of the voltage divider. The SSU was built on the interdigitated electrodes and had a good linearity characteristic between its resistance and hand sweatiness, which was simulated through different masses of normal saline. By applying an AC power supply, the crosstalk between the SSU and the FSU was significantly attenuated. Two active electrodes and one reference electrode were exploited in the EMGSU, and by designing the particular shapes of the electrodes and locating them properly, the EMG signals were acquired robustly with the help of an instrumental amplifier. This smart sensor platform with promising sensing capabilities can be used within a large number of applications that involve continuous, real-time physical and psychological condition monitoring for drivers, including health monitoring, cognitive assessment, and driver assistance.

## Figures and Tables

**Figure 1 sensors-22-07296-f001:**
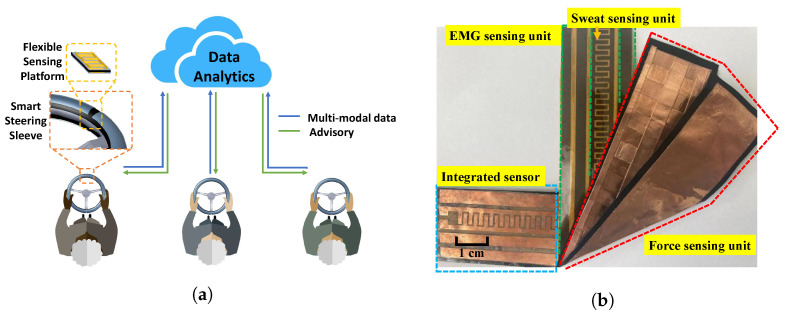
Overview of the developed cloud-based smart steering sleeve system: (**a**) schematic of the sensor and cloud-based data analytics; and (**b**) integrated multi-modal sensor prototype.

**Figure 2 sensors-22-07296-f002:**
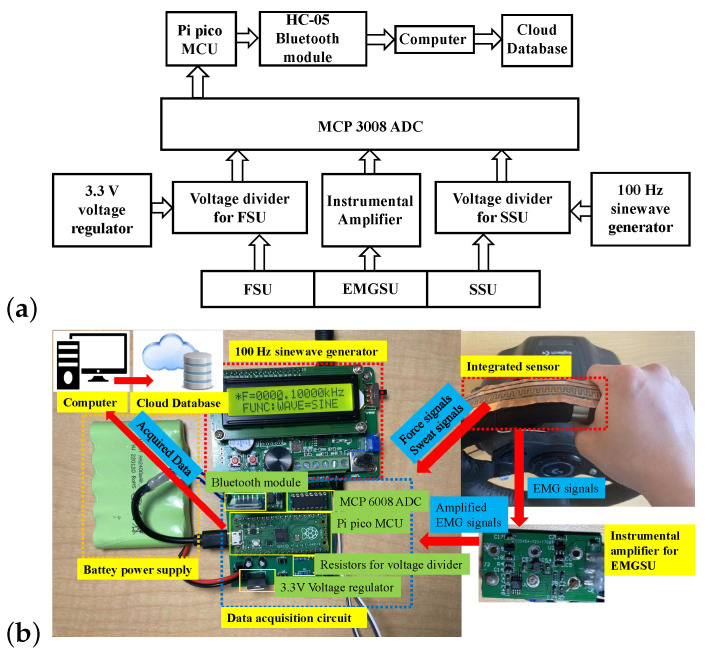
The measurement system for the integrated sensor: (**a**) diagram of the measurement system; and (**b**) the prototype of the measurement system.

**Figure 3 sensors-22-07296-f003:**
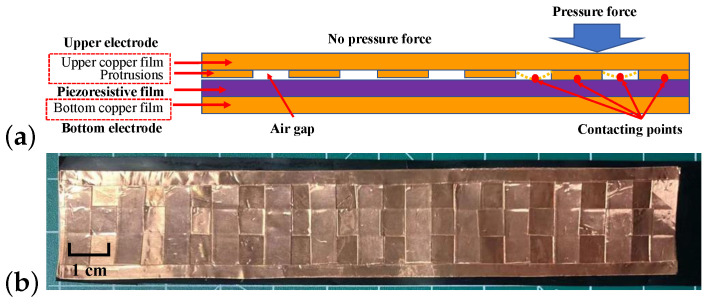
The design and the mechanism of the FSU: (**a**) comparison between with and without applied force on the FSU; and (**b**) prototype of cross-shape pattern in the FSU for creating protrusions.

**Figure 4 sensors-22-07296-f004:**
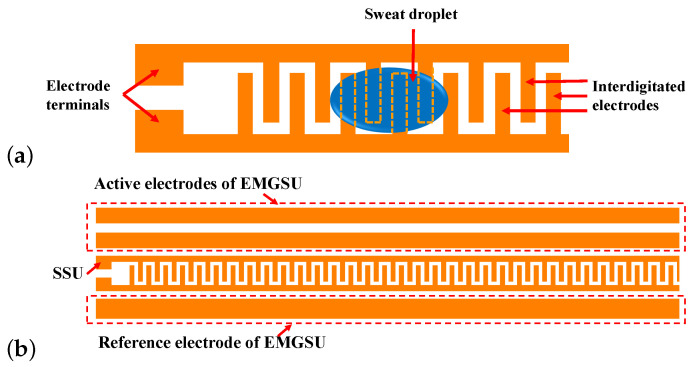
The design and mechanism of the SSU and EMGSU: (**a**) schematic of the design and mechanism of the SSU; and (**b**) illustration of the EMGSU and the SSU.

**Figure 5 sensors-22-07296-f005:**
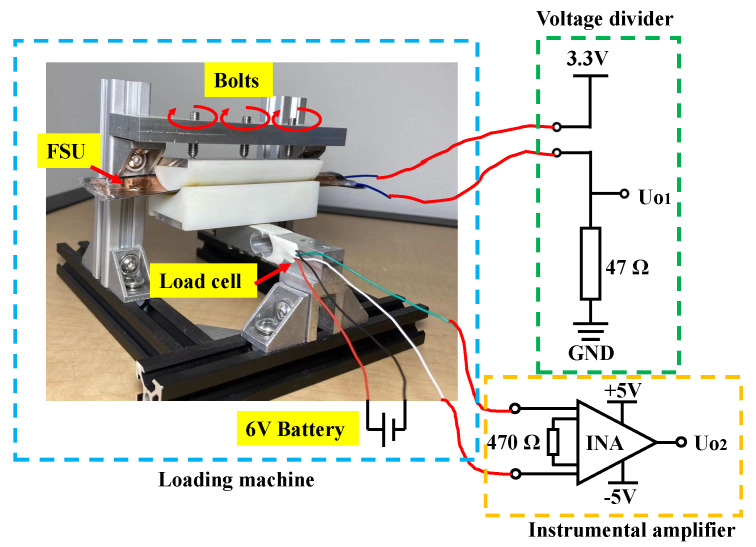
The calibration platform for the FSU.

**Figure 6 sensors-22-07296-f006:**
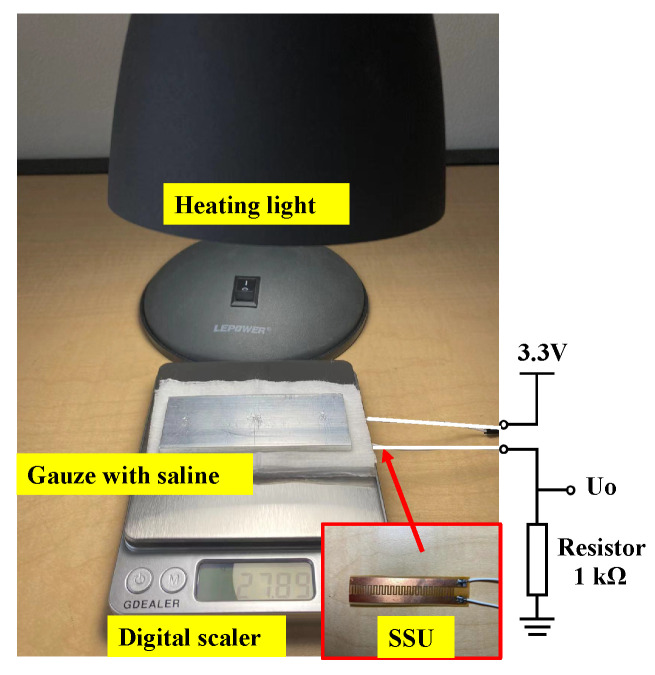
Devices used for resistance variation measurement of the SSU.

**Figure 7 sensors-22-07296-f007:**
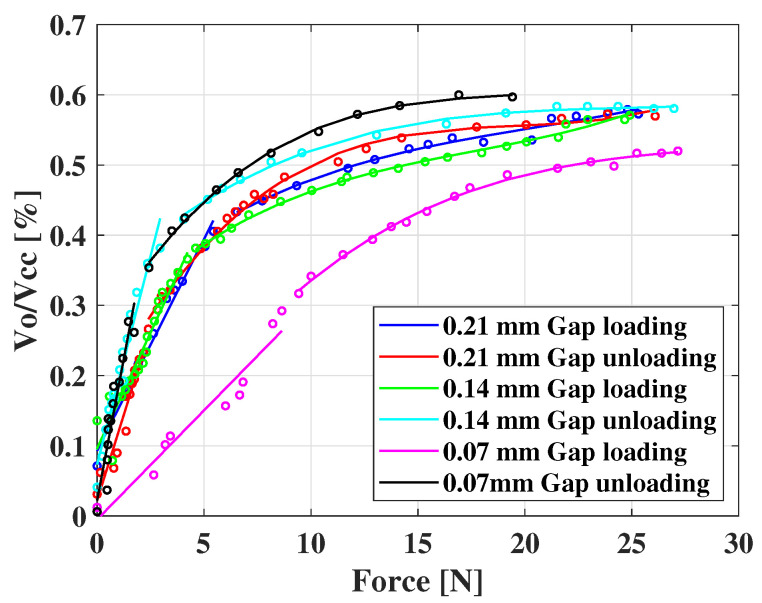
The calibration results of the FSUs with different numbers of layers.

**Figure 8 sensors-22-07296-f008:**
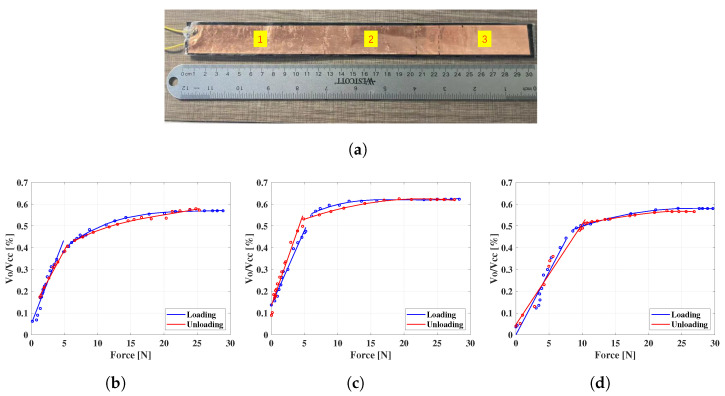
The sensitivity curves of the three parts of the FSU: (**a**) the location of the left, middle, and right portions of the FSU under the test of sensitivity curves; the curves from the (**b**) left part; (**c**) middle part; and (**d**) right part. After the load is more than approximately 25 N, the output ratio of the voltage divider Vo/Vcc is saturated.

**Figure 9 sensors-22-07296-f009:**
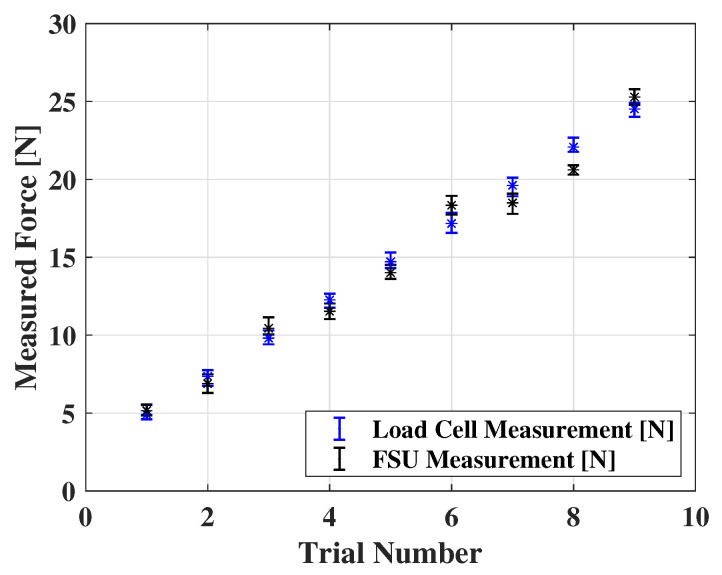
The comparison between a commercial load cell and our FSU under various force loads.

**Figure 10 sensors-22-07296-f010:**
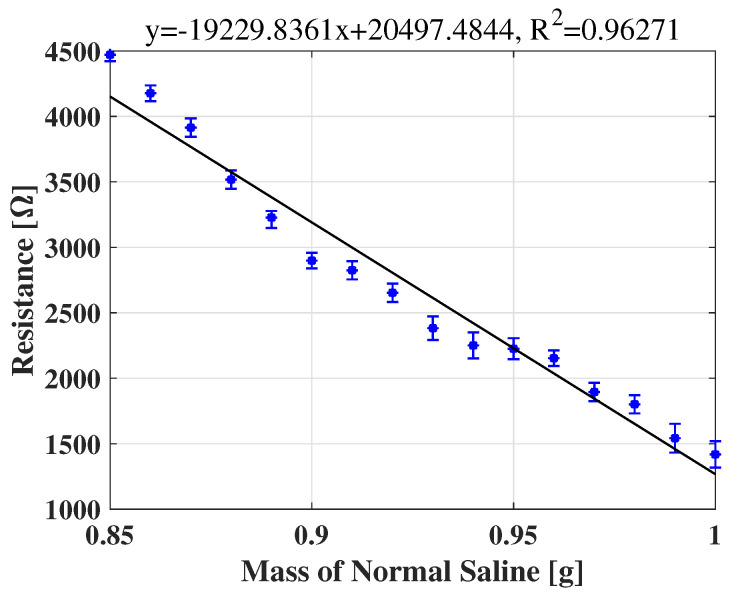
The relationship between the mass of the normal saline and the resistance of the SSU.

**Figure 11 sensors-22-07296-f011:**
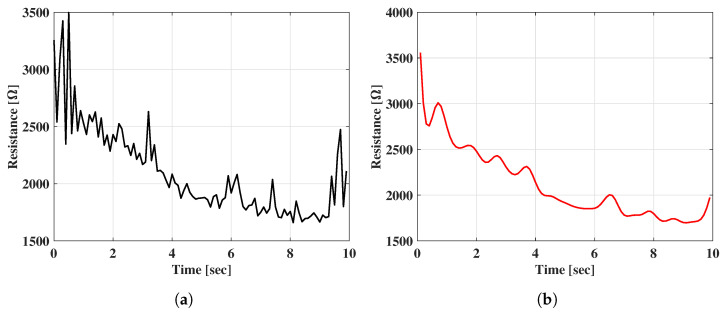
The resistance curves of the SSU when adding the normal saline: (**a**) resistance curve under the varying force applied; and (**b**) resistance curve after the Butterworth filter applied.

**Figure 12 sensors-22-07296-f012:**
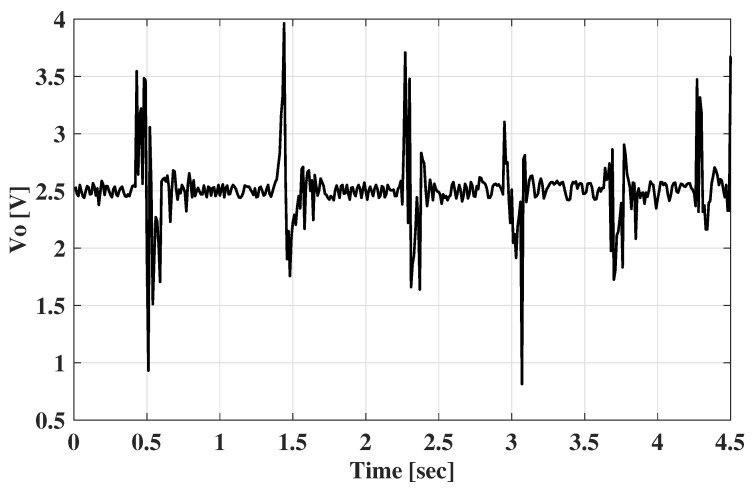
The amplified EMG signals acquired by the DAQ device.

**Figure 13 sensors-22-07296-f013:**
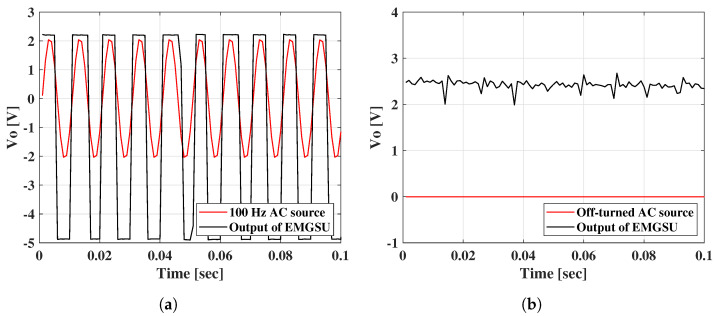
The solution used to eliminate the crosstalk between SSU and EMGSU: (**a**) an oscillation occurred in EMGSU channel when the 100 Hz AC source for the SSU was turned on; and (**b**) the EMG signals were picked up effectively by the EMGSU when the AC source was turned off.

**Figure 14 sensors-22-07296-f014:**
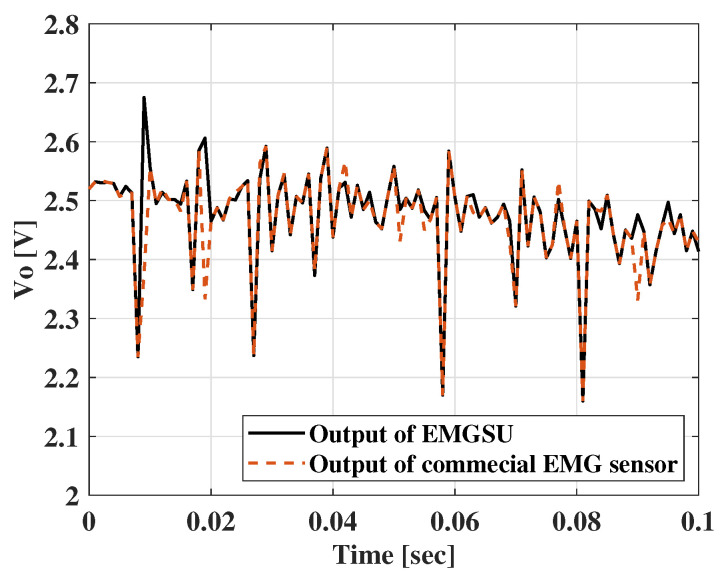
Comparison of EMG measurements between a commercial EMG sensor and our EMGSU.

## Data Availability

Not applicable.

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
