# Peer review of "Smart Steering Sleeve (S3): A Non-Intrusive and Integrative Sensing Platform for Driver Physiological Monitoring"

_sensors, 2022, doi:10.3390/s22197296_

Round 1

Reviewer 1 Report

Paper is well written, but the scientific contributions to knowledge should be improved. What are the specific contributions to knowledge with reference to state-of-the-art. What has the research added to knowledge that is novel-doing things differently, etc?

Typos should be checked and corrected, e.g. in Abstract: replace 'Driving is a ubiquitous activity' with 'Driving is an ubiquitous activity'. Section 5, replace 'it can potentially used' with 'it can potentially be used'

Performance of numerical or simulated results should be mentioned in the Abstract.

Any implications of the study or possible effects of the sleeve on health or environment?

Any comparison of design features with state-of-the-art? Could not clearly distinguish the gap(s) between this research and state-of-the-art.

Author Response

Dear Reviewer,

Thank you for your comments on the manuscript, and we have revised it according to your suggestions thoroughly. Please see the attachment.

Thanks again for your effort!

Please refer to the response to Reviewer 1 in the attached PDF file.

Reviewer 2 Report

This paper proposes a multi-modal smart steering sleeve system that can non-intrusively and continuously measure a driver's physiological signals. Firstly, a system capable of monitoring the driver's force, palm sweat and electromyography (EMG) in real time is assembled on a tape-like strip attached to a steering sleeve. Meanwhile, the data detected by the system could be managed and analyzed through a cloud-based communication architecture. Then, mechanisms and devices are designed and developed for accurate and robust characterization of the three sensing units. Finally, comprehensive characterizations are performed on the integrated sensor platform with the results demonstrated. Generally, this paper is well-written and easy to follow. The proposed system is very practical and can significantly reduce production costs. However, there are still some suggestions for authors’ references. Below are my comments:

1.     The purpose of this paper is to provide uninterrupted real-time monitoring of the driver's state, and from a practical point of view, the output of the system should be able to determine a certain aspect of the driver's state and thus assess driving behavior. However, the paper only mentions this in the introduction and application, and the methods section fails to correspond the results of force, palm sweat and EMG to the results of the driver's physiological state monitoring, which may make the paper less valuable.

2.     On the basis of the above-mentioned unknown results of driver physiological monitoring, the accuracy of the results obtained in this article for force, palm sweat and EMG is also unsupported. Moreover, the paper does not propose an effective validation method to evaluate the accuracy of the system. It is known that changes in driver status can be extremely small when reflected in the three indicators described in the paper, so the accuracy of the results is particularly important at this time.

3.     Both indicators, pressure and hand sweat, are external manifestations and are closely related to driver habits, strength, and whether or not they sweat easily, which varies from person to person. For example, some drivers are more sensitive and variable in their grip strength performance, but this does not mean that they are more prone to tension than other drivers. Similarly, the factor of sweaty hands is even more person-specific, and the fact that drivers with sweat-prone physiques perform differently than drivers with the opposite physique does not indicate that the former are experiencing more thrilling driving situations than the latter. Therefore, the system needs a lot of theoretical support when applied to driver condition monitoring.

4.     In the third point of the contribution the authors mention the comprehensive characteristics, while the latter article does not mention the comprehensive characteristics after modeling the three indicators separately. In addition, do the three indicators described in the paper affect each other? For example, does more sweaty hands reduce force? Can increased pressure diminish the sensitivity of the EMG? How is the combined effect of the three modeled? Is there a uniform scale for all three?

5.     The system could provide a more cost-effective solution for industry. However, in the context of scientific research, it is less innovative in the field of biomaterials and less cutting-edge in the field of driver behavior research, where the relationship between physiological indicators and traffic indicators has still not been built.

Author Response

Dear Reviewer,

Thank you for your comments on the manuscript, and we have revised it according to your suggestions thoroughly. Please see the attachment.

Thanks again for your effort!

Please refer to the response to Reviewer 2 in the attached PDF file.

Reviewer 3 Report

Chuwei Ye et al., report on the development of a steering sleeve system that can monitor driver physiological signals, comprising sweat loss (SSU), electromyography (EMGSU), and hand force (FSU). The topic is interesting and can be appealing for a broad readership. However, only calibration tests have been performed and the work lacks a test on drivers. Additionally, the manuscript contains some inconsistences and data are not always clearly commented, therefore I suggest publication only after revision.

In detail, my comments are as follow:

1       Introduction: in general, the novelty and the originality of the work should be more stressed. Furthermore, the introduction could be extended to better focus the state of art (see also comment n2).

2       Page 2, lines 43-46. The reviewer agrees with the authors on the fact that wearable sensors can “reduce the acceptance and adoption rate among the drivers”. Nevertheless, authors omitted to mention and cite published works that report on non-wearable technologies for driver physiological monitoring; see for instance:

- M. Gjoreski, et al., IEEE Access 2020, 8, 70590-70603. doi: 10.1109/ACCESS.2020.2986810.

- Minea, M., et al., Sensors 2021, 21(24), 8272. https://doi.org/10.3390/
s21248272.

- H. J. Baek, et al., Telemed J E Health. 2009 Mar;15(2):182-9. doi: 10.1089/tmj.2008.0090.

- S. Leonhardt, et al., Sensors 2018, vol. 18, no. 9, pp. 1–38, 2018, issn: 14248220. doi: 10.3390/s18093080.

- H. Qi, et al., IEEE China Summit and International Conference on Signal and Information Processing (ChinaSIP), 2015, pp. 418-422, doi: 10.1109/ChinaSIP.2015.7230436.

- T. Wartzek, et al., Biomedical Engineering, IEEE Transactions on, vol. 58, no. 11, pp. 3112–3120, 2011. DOI: 10.1109/TBME.2011.2163715

- R. J. Van Loon, et al., Accident Analysis and Prevention, vol. 84, pp. 134–143, Nov. 2015, issn: 00014575. doi: 10.1016/j.aap.2015.08.012.

These works should be added. Furthermore, a benchmarking with the system presented in this paper could be as well carried out, later in the article.

3       For the behalf of clearness and easiness of reading, some figures could be merged; for instance, Figures 4 and 5, Figures 10 and 11, and Figure 2b and Figure 8, which report the same picture.

4       The reported FSU system has been tested in a range between 0 N and 25 N (Figures 9 and 11). According to literature, the force used by a driver could reach values of hundreds of Newtons (see for instance Eksioglu, M., et al., Steering-wheel grip force characteristics of drivers as a function of gender, speed, and road condition. International journal of industrial ergonomics 2008, 38(3-4), 354-361. Rahayu, S., et al., A comparison study between right hand and left-hand grip pressure force while driving. Australian Journal of Basic and Applied Science 2015, 9(19), 50-58.) Authors should add measurements at higher force values or at least address this issue in the text and discuss their choice of a limited force range.

5       Right- and left-hand grip force values are often very different while driving. Furthermore, grip force while driving depends on several parameters, including the gender of the driver, the speed of the car and the condition of the road. Authors should address these issues and discuss them in the text.

6       Page 8, lines 234-236 and Figures 9 and 11: what is the reason beyond the two different slopes in the calibration curve? Are the authors sure that for higher force values (which are the values related to real driving) another change in slope does not occur? A comment should be added in the text.

7       Page 8, line 242: authors wrote that ref [23] assessed that the mass variation of sweat released by drivers is less than 8 mg. However, ref [23] does not report on drivers and the reviewer cannot find the information about the 8 mg of sweat. Please, check if the reference is corrected.

8       Figure 12: error bars should be added to the resistance value. Furthermore, the tested saline mass range seems a bit limited, considering the 8 mg value aforehand mentioned by the authors themselves. Finally, the relation between resistance and saline mass seems more like a power law rather than a line. I suggest adding more data, also to completely rule out the power-law dependence or the line trend.

9       The conclusion section should be extended and improved to help a reader not familiar with the topic.

10    Carefully check English through the whole manuscript.

11    Finally, the paper presented only a calibration for the FSU, SSU and EMGSU, and, especially in the case of the FSU, the calibration range is not enough to prove the capability of the device to real-time driver monitoring. Tests of the device on volunteers (in real driving condition or at least in lab) should be carried out in order to prove the effective capability of the device to be applied for driver physiological monitoring.

Author Response

Dear Reviewer,

Thank you for your comments on the manuscript, and we have revised it according to your suggestions thoroughly. Please see the attachment.

Thanks again for your effort!

Please refer to the response to Reviewer 3 in the attached PDF file.

Round 2

Reviewer 3 Report

The authors carefully revised the whole manuscript, according to the suggestions made by all the referees. The state of art is now clearly discussed in the introduction, as well as the novelty of the work. The manuscript has been improved both in quality of the results and clarity in their presentation and discussion. Nevertheless, before publication I still have a comment for the authors: regarding the calibration curves and the linear fitting reported in Figure 7 and Figure 8, although a linear fit is appropriate for the low force range (approximately 0-5N), it does not seem the right choice of the remaining range. I suggest considering a linear fit for the first part and then a power law fit, which will be better suited especially for the saturation trend.

Author Response

Hi editor and reviewer,

Thank you for your comments about my manuscript, and the revision is attached.

Best regards.
